# Development of a Meal-Planning Exchange List for Traditional Sweets and Appetizers in Gulf Cooperation Council Countries: Insights from Qatar

**DOI:** 10.3390/nu18010117

**Published:** 2025-12-29

**Authors:** Safa Abdul Majeed, Reema Tayyem

**Affiliations:** Department of Nutrition Sciences, College of Health Science, Qatar University, Doha P.O. Box 2713, Qatar

**Keywords:** sweets, appetizer, food exchange list, GCC cuisine, cultural foods, Qatari cuisine

## Abstract

Background & Aim: Sweets and appetizers are an integral part of the Gulf Cooperation Council (GCC) region’s cultural heritage but are often high in refined carbohydrates, sugars, and fats, contributing to the rising burden of obesity and type 2 diabetes. Qatar, as one of the fastest-developing GCC nations, exemplifies these nutrition-related challenges. Therefore, this study aimed to systematically develop a culturally adapted meal-planning exchange list for 34 commonly consumed Qatari and GCC sweets and appetizers to support nutrition counseling and diabetes management. Methods: This study is primarily methodological and developmental in scope, employing a descriptive observational design in which the units of analysis were the 34 selected traditional dishes. Standardized recipes were compiled for each dish, and serving sizes were determined. Macronutrient content (carbohydrates, protein, fat) was analyzed, variability across dishes was assessed, and nutrient data were validated against food processor software data. Results: The nutrient comparison analysis revealed strong correlations between collected nutrient data sources (r = 0.81–0.85, *p* < 0.05) and significant variability in macronutrient profiles. Fried and sugar-syrup-based items presented higher fat and carbohydrate content, while legume- and vegetable-based dishes contributed additional protein and fiber, demonstrating the dual role of traditional foods as both nutrient rich and energy dense. Conclusions: The developed exchange list provides a practical tool for culturally relevant nutrition guidance. It enables dietitians to plan individualized meals, promoting moderation, portion control, and adherence to dietary recommendations, thereby supporting diabetes and weight management initiatives across the GCC.

## 1. Introduction

Traditional sweets and appetizers occupy a central role in Arab culture, serving as profound expressions of hospitality, heritage, and communal joy [1]. Across the Gulf Cooperation Council (GCC) countries, including Qatar, Saudi Arabia, the United Arab Emirates, Kuwait, Bahrain, and Oman, sweets such as *maamoul*, *luqaimat*, and *khanfaroosh* are not merely desserts but symbols of warmth, generosity, and shared identity during events like Ramadan, Eid, weddings, and family gatherings [2].

At the same time, GCC countries face the world’s highest rates of obesity and type 2 diabetes mellitus. The International Diabetes Federation (2023) reported that more than 1 in 5 adults in the Gulf region lives with diabetes, with prevalence rates exceeding 15% in several member states [3]. Qatar mirrors these regional trends, with type 2 diabetes affecting 230 per 1000 Qataris and 183 per 1000 non-Qataris, with prevalence projected to rise by at least 24% by 2050 [4,5]. These high rates are largely linked to obesity, which is strongly influenced by unhealthy diets rich in refined carbohydrates, added sugars, and saturated fats, alongside sedentary lifestyles [6,7].

Recent research across the GCC highlights excessive sugar and fat consumption, particularly among Qatari adults, with mean daily sugar intakes reaching 153 g/day and over 70% of adults classified as overweight or obese [8]. An analysis of dietary patterns using Qatar Biobank data from 1893 adults identified a “sweet dietary pattern” characterized by high intake of traditional desserts and sugar-sweetened beverages [9]. While this study found no significant association between the sweet dietary pattern and diabetes prevalence in their clinical sample, the broader regional evidence suggests concerning links between energy-dense traditional foods and cardiometabolic disorders [6,7]. Similarly, the Qatar stepwise national survey (*n* = 1109) reported that 50–72% of participants consumed sweetened beverages and sweets frequently [10]. Traditional sweets, often prepared with refined sugars, clarified butter (samn), nuts, and dates, are calorie-dense with high glycemic potential. Their cultural context, typically consumed in large portions during festive occasions and social gatherings, further amplifies their contribution to excessive carbohydrate and saturated fat intake, thereby fueling the nation’s rising burden of metabolic syndrome, cardiovascular disease, and diabetes complications [6,9].

Addressing these challenges requires the development of culturally relevant meal-planning exchange lists, given that ethnic variations and traditional practices have a profound influence on food choices [11]. Therefore, incorporating culturally accepted foods into dietary planning is essential for promoting adherence to healthy eating. In addition, this list serves as a practical tool that enables individuals, particularly those with diabetes or metabolic disorders, to make informed food choices by providing nutritionally equivalent portions of commonly consumed foods [12]. They also facilitate carbohydrate counting, portion control, and structured meal planning, which are key components of effective diabetes management and weight control delivered by nutrition health professionals [13].

Evidence further indicates that culturally tailored dietary planning tools play a critical role in supporting adherence to medical nutrition therapy (MNT) and improving outcomes in non-communicable disease (NCD) management. Studies in diabetes and cardiovascular disease have demonstrated that when dietary recommendations align with traditional foods and regional eating patterns, patient compliance, dietary satisfaction, and long-term metabolic outcomes improve substantially [14,15]. However, existing food exchange lists in Qatar do not adequately capture the wide variety of traditional and common sweets and appetizers, creating barriers to dietary adherence and limiting the effectiveness of nutrition counseling. The Transcultural Diabetes Nutrition Algorithm for the Middle East underscores that medical nutrition therapy achieves superior outcomes when adapted to regional foods and cultural practices [7].

Despite the cultural and dietary significance of traditional sweets and appetizers in Arab communities, no exchange lists have been developed for these foods in the Gulf region. As a result, healthcare providers lack culturally appropriate tools for delivering accurate nutritional guidance.

Therefore, the primary objective of this study is to develop a culturally relevant meal-planning exchange list for 34 traditional sweets and appetizers commonly consumed in Qatar and the Gulf region. By establishing standardized exchange values for these foods, this tool intends to support dietitians and individuals in making informed dietary decisions and to strengthen nutrition management strategies for the growing burden of diet-related non-communicable diseases (NCDs) in the region.

## 2. Materials and Methods

This study is primarily methodological and developmental in scope; however, it also follows a descriptive observational design, where the units of analysis were 34 commonly consumed traditional Qatari/GCC sweets and appetizers. A structured, multi-phase framework was employed to ensure both scientific rigor and cultural relevance in the development of the food exchange list. The methodological workflow used to develop the exchange list is illustrated in Figure 1, which outlines the four sequential phases applied to all analyzed dishes (*n* = 34).

### 2.1. Phase 1: Selection of the Dishes

Dishes were selected based on the GCC and Qatari food culture, culinary practices, and eating habits, as well as the availability of these foods in restaurants and local markets [16]. This approach ensures that both traditional dishes that reflect GCC’s and Qatar’s cultural heritage and popular foods that are part of the modern diet are captured.

Dishes were included if they were traditional, consumed as a sweet or appetizer, and commonly reported in regional food consumption studies. The following criteria were used for the selection:Frequently consumed in GCC and Qatar, either daily (e.g., breads, dips, appetizers) or during festive occasions such as Ramadan, Eid, weddings, and family gatherings (e.g., traditional sweets).Held cultural or traditional significance, forming part of GCC and Qatari heritage and social practices.Widely available in households, restaurants, or bakeries, ensuring accessibility across settings.Contributed substantially to carbohydrate, fat, or overall energy intake in the GCC and Qatari diet [17].Represented distinct preparation categories (e.g., fried sweets, puddings, bread-based items, savory appetizers).Items with highly variable recipes or limited regional consumption were excluded to ensure standardization feasibility.The selected dishes represent those with the greatest relevance for Medical Nutrition Therapy, due to their high contribution to carbohydrate, fat, or sugar intake, especially among individuals with diabetes and cardiometabolic risk.

The selection process considered the types of establishments, including traditional family-run diners, modern casual dining outlets, and neighborhood markets (souqs). Traditional markets and household preparations typically reflect authentic GCC and Qatari cuisine, using local ingredients such as dates, samn (clarified butter), wheat, legumes, and aromatic spices. Traditional sweets often feature floral, nutty, and buttery flavors, while appetizers highlight spiced, aromatic, and slow-cooked profiles, both representing GCC and Qatar’s culinary heritage that blends the simplicity of its Bedouin desert traditions with rich influences from historical maritime trade across the Arabian Gulf. In contrast, modern restaurants offer modified versions of these dishes, often influenced by Western cooking styles and ingredient substitutions, such as the incorporation of cheese, chocolate, cream, and European-style pastry techniques. This illustrates the coexistence of culinary authenticity and modernization within the GCC and Qatari food landscape.

This selection approach ensured representation of dishes accessible to different socioeconomic groups, ranging from affordable, everyday meals to those typically served during festive or special occasions.

A total of 34 traditional and popular dishes (22 sweets and 12 savory appetizers) that met the inclusion criteria were selected, and both the number of dishes and the criteria were maintained throughout all four phases of the development process. The sample size (*n* = 34) was based on a list of frequently consumed dishes identified from regional surveys and validated by local dietitians, rather than a pre-calculated power analysis. Due to the extensive variety of traditional sweets and appetizers, quantifying the proportion of all regional dishes represented by the selected samples is challenging, as comprehensive documentation is lacking in published literature and national food composition tables. To ensure cultural relevance and practical applicability, we conducted dietary surveys to assess the frequency of consumption and market availability of various traditional foods, complemented by insights from local dietitians. The final selection of 34 dishes was based on their cultural significance and consumption patterns across different socioeconomic groups. This core group reflects the most meaningful dishes for dietary counseling, rather than exhaustively cataloging all traditional foods in the region.

Even though westernization influenced the preparation of dishes, the core ingredients remain the same. Most variations in preparation among households and restaurants occur in the use of additional minor ingredients (such as nuts, spices, or sweeteners), the type of oil or fat used, and the specific cooking method. Variations in the type of oil or fat used do not substantially alter macronutrient content because all culinary oils provide approximately 9 kcal per gram of fat and contain similar total fat content. Likewise, substitutions among commonly used sweeteners (e.g., white sugar, brown sugar, date syrup) result in minimal differences in the total carbohydrate content, as these sweeteners contribute comparable grams of carbohydrates per serving. To further enhance reliability and reduce this inherent variability, a standardized recipe development process was implemented. Initially, recipes were gathered from numerous sources, which included published research on Qatari cuisine [18,19], food composition tables from national and regional sources (Bahrain [20], Jordan [21], EMFID [22]), and traditional cookbooks [16]. When multiple recipe versions or laboratory-analyzed nutrient values for the same dish were available across different sources, the version that appeared most frequently across sources was selected as the standardized form (e.g., if a recipe or nutrient profile was reported similarly in two out of three sources, that version was chosen). While this standardized approach ensures consistency for analysis, it may not fully capture real-world household or regional variations in preparation. Major ingredients and cooking methods consistently cited across three or more sources, with a similarity of ≥75% across these sources, were retained. Optional components, such as nuts or garnishes, were omitted unless they were widely used. Cooking techniques were standardized according to the most prevalent traditional and common practices for each dish (for instance, fried versus baked variations). Finally, the selected recipes and nutrient profiles were validated against ESHA (Electronic Software for Health Analysis) Food Processor (version 11.11.0; ESHA, Salem, OR, USA) to ensure consistency and cultural relevance. A comprehensive list of ingredients for these dishes is provided in Appendix A Table A1.

Differences in household portion sizes also exist; however, the study standardized portion sizes based on the most commonly available commercial or culturally recognized serving units to minimize this variation.

### 2.2. Phase 2: Collection of Food Composition Data for Selected Dishes

Direct laboratory analysis of the selected traditional dishes was not conducted as part of this study. Instead, nutrient composition values were obtained from published, laboratory-analyzed food composition studies to ensure scientific accuracy. For each dish, we systematically reviewed nutrient data from multiple authoritative regional food composition tables (e.g., Bahrain, Jordan, EMFID, GCC sources). The nutrient values reported in these tables are themselves derived from standardized laboratory analyses using established analytical techniques, which provides a credible foundation for the exchange list.

To minimize discrepancies, nutrient profiles for each dish were compared across all available sources. When more than one laboratory-derived value was available, the most consistent and frequently occurring value was selected. For instance, if the carbohydrate content of a dish such as Luqemat was reported as 28 g in three separate food composition references, the repeated value was chosen as the representative figure. This cross-source consistency approach reduces the potential for random variation and enhances the stability of the selected nutrient estimates.

Following selection, all nutrient values were cross-validated using ESHA Food Processor SQL (version 10.1.129; ESHA, Salem, OR, USA) and additional regional references to confirm alignment with culturally appropriate recipes and commonly used ingredients. This multi-step verification ensured that the nutrient profiles reflect widely consumed, standardized commercial or home-style preparations, rather than uncommon or idiosyncratic variants.

Due to the lack of comprehensive and updated food composition tables or nutritional databases specific to Qatar and Gulf countries, four main external sources are used. Among the total 34 dishes, the nutrient content of 4 dishes was selected from Qatar food composition studies conducted by El Obeid [18], and Nagdy et al. [19], and nutrient data for 9 dishes were drawn from the food composition table of Bahrain [20]. For the remaining 21 dishes, the nutrient content was taken from the Jordanian food composition table [21]. By relying on laboratory-analyzed data from multiple harmonized regional sources and applying a consistency-based selection strategy, this methodology ensures methodological rigor, despite the absence of direct laboratory testing. It minimizes variation arising from household-level recipe differences, cooking methods, or ingredient substitutions and enhances reproducibility by grounding the selected nutrient profiles in widely recognized peer-reviewed regional food composition tables. This approach strengthens the overall validity of the exchange list by ensuring that the nutrient estimates accurately reflect foods commonly consumed in the Gulf and Middle East region.

#### Statistical Analysis

The macronutrient data (carbohydrate, protein, and fat per 100 g edible portion) for the 34 selected traditional Qatari/GCC sweets and appetizers were obtained from multiple sources. Table 1 lists the source of nutrient data for each dish and the corresponding preparation type (domestically prepared or commercially purchased). For each dish, macronutrient values (total carbohydrates, protein, and total fat, g/100 g) were extracted and compiled into a single dataset. The compiled nutrient data were then validated by comparison with values generated using dietary analysis software (Food Processor SQL version 10.1.129; ESHA, Salem, OR, USA) [23]. All macronutrient variables were treated as continuous variables. We used Pearson’s correlation coefficient (r) for normally distributed data, while Spearman’s correlation coefficient (ρ) was used for data that were not normally distributed. The statistical evaluation was performed using Stata SE 18 (Stata Corp LLC, College Station, TX, USA). Significance level was set as *p* value < 0.05.

### 2.3. Phase 3: Determination of Serving Size

All dishes were first standardized to 100 g of edible portion. From this, the serving size that best aligned with the Wheeler et al. method [24] was identified and then adjusted to match portion sizes commonly available in local stores and hypermarkets. The researchers converted the selected standard weights into practical household measurements using a calibrated kitchen scale (Model F1976:14191-38B, China) and standard measuring cups. The resulting household measures were then rounded and validated by another local dietitian and further compared with the portion sizes and packaging most frequently found in hypermarkets and food stores. This comprehensive, researcher-led process ensured that the final exchange portions were both culturally appropriate and reflective of traditional serving practices within Qatari and broader GCC meal patterns. Figure 2 shows the steps of portion size estimation.

Once the proximate analysis of each dish was obtained, nutrient values per serving were calculated by dividing the carbohydrate, protein, and fat content of the 100 g sample by the number of servings. For example, Memolo bil fustok (cookie filled with pistachio) contains 16.7 g carbohydrate, 5.5 g protein, and 7.7 g fat per 100 g. In retail stores, these cookies are typically sold in 35 g packs, which is therefore considered one serving. Accordingly, one serving provides approximately 6 g of carbohydrate, 2 g of protein, and 2.6 g of fat.

### 2.4. Phase 4: Fitting the Dishes in the Exchange List

Finally, a macronutrient exchange list for each dish was developed using the rounding-off criteria established by Wheeler and colleagues [24]. The adopted Wheeler exchange principles align with internationally used systems such as the Academy of Nutrition and Dietetics’ food exchange system and the American diabetes association framework, ensuring cross-system compatibility.

The exchange values were calculated based on the collected food composition data (Table 2) as follows:Carbohydrate exchange:

A dish does not count as a serving if it has between 1 and 5 g of carbohydrates. If the food has 6–10 g of carbohydrates, it is considered half a serving, and if it contains 11–20 g of carbohydrates, it counts as one serving.

Protein exchange:

For meat and meat substitute dishes, an amount of protein between 0 and 3 g is not counted as a serving. If there are 4–10 g of protein, it counts as one serving.

Fat exchange:

If a food portion has between 0 and 2 g of fat, it is not considered a serving. If it contains 3 g of fat, it is counted as half a serving, and if there are 4–7 g of fat, it is considered one serving.

**Table 2 nutrients-18-00117-t002:** Standardized macronutrient cut-offs used for calculating exchange servings across dishes.

**Carbohydrate exchange:**
1–5 g carbohydrate → not counted 6–10 g carbohydrate → ½ serving 11–20 g carbohydrate → 1 serving
**Protein exchange (meat and substitutes):**
0–3 g protein → not counted 4–10 g protein → 1 serving
**Fat exchange:**
0–2 g fat → not counted 3 g fat → ½ serving 4–7 g fat → 1 serving

## 3. Results

The nutritional analysis covered 34 traditional Qatari and GCC sweets and appetizers.

The correlation analysis of collected macronutrient values (total carbohydrates, protein, and total fat, expressed in grams per 100 g edible portion) obtained from various food composition tables and the corresponding values generated by ESHA’s Food Processor for each dish category indicated a notable and significant correlation (*p* < 0.05). The correlation coefficients and corresponding *p*-values are presented in Table 3. Pearson’s correlation coefficient (r) was used for all macronutrient variables, as the data were normally distributed. It should be noted that this validation was performed through software comparison and cross-reference with food composition tables, and not through direct laboratory analysis. Thus, while the correlations indicate good agreement, some variability in real-world preparation may still exist.

Among the dish groups, appetizers exhibited the strongest correlations for carbohydrates (r = 0.90) and protein (r = 0.92), whereas the correlation for fat was comparatively weaker (r = 0.53). In contrast, sweets demonstrated a higher correlation with fat (r = 0.80). Overall, these findings suggest a good level of agreement between the two data sources for nutrient estimation; however, greater variability in fat content estimation was noted for appetizer dishes, likely due to differences in sesame or oil quantities across households and commercial preparations.

Table 4 and Table 5 present the finalized culturally adapted food exchange list, detailing their serving sizes, macronutrient content, and the calculated food exchanges for each.

Considerable variation was observed in the macronutrient composition of traditional desserts (*n* = 22). Fat content showed the greatest variability, ranging from 0 to 35.8 g per 100 g, with Aigalee and Assedah being notably high in fat, while Barinoish and Sago contained minimal fat. Protein-rich desserts, such as Konafah na’ema bil jibn and Khanfarooshe, reflected the inclusion of eggs and cheese, adding milk or meat exchanges. Preparation methods, such as frying versus grilling, also influenced fat content, as seen in variations of Qatayif. On average, these energy-dense sweets contained 31.5 g carbohydrates, 4.0 g protein, and 8.5 g fat per serving, contributing predominantly starch, fat, and other carbohydrate exchanges, with 1–6 exchanges per serving. These findings underscore the heterogeneity of traditional desserts and the need for careful portion control and individualized dietary exchange planning.

Appetizers (*n* = 12) centered on legumes, vegetables, grains/bread, and sesame paste, exhibited considerable variation in macronutrient composition. Carbohydrate content ranged from 4.5 to 27.1 g per 100 g, protein from 1.3 to 11.1 g, and fat from 4.9 to 19.6 g per 100 g. Dishes prepared with sesame paste, such as Hummus bil tahina, Baqdonsieh, and Motabbal bathinjan bil tahina, showed higher fat values and contributed multiple fat exchanges. Legume-based items, including Falafel and Foul modammas, provided the highest protein content and contributed both starch and low-fat meat exchanges. Vegetable-rich salads, such as Tabbouleh and Fatoosh, were lower in carbohydrates and mainly contributed to vegetable exchanges with moderate fat from olive oil. Fried dishes, including Sambosik bil sabanikh, Falafel, and Qatayif maqli, had higher fat content compared with roasted or blended preparations, highlighting the influence of cooking method on nutrient composition and exchange classification. On average, appetizers contain 13.5 g carbohydrates, 5.8 g protein, and 10.2 g fat per serving, reflecting their higher protein contribution from legumes and moderate fat from sesame and oils.

All final portion sizes were aligned with both the Wheeler et al. [25] exchange standards and the Qatar Dietary Guidelines to ensure consistency with internationally recognized exchange systems and national dietary recommendations. While a direct per-dish comparison with these guidelines is not feasible due to the composite nature of traditional sweets and appetizers, the standardized serving sizes were derived from mean nutrient profiles and rounded to fit within the QDG and international exchange thresholds for specific food groups (Starch, Protein, Fat). The comparison of the mean calculated GCC exchange profile with the international standards is presented in Table 6.

## 4. Discussion

This study presents the first systematic development of a culturally adapted meal-planning exchange list for 34 traditional Qatari and GCC sweets and appetizers. This novel work fills a critical gap in regional dietary resources, providing nutrition professionals with a tool grounded in local cuisine to enhance the effectiveness of nutrition counseling for diet-related NCDs. It extends the concept of exchange lists by contextualizing it within the Gulf region’s food culture, where mixed dishes and traditional meal patterns require unique categorization strategies. By bridging cultural dietary practices with standardized nutritional equivalence, the present work contributes to the localization of global dietary tools for practical use in Arab populations.

### 4.1. Cultural and Public-Health Significance of GCC Sweets and Appetizers

The inclusion of both home-style and restaurant-prepared dishes, spanning a range of socioeconomic accessibility, reflects the diversity of the local diet. Traditional staples such as Foul Modammas, Mutabbal, Hummus, Tabbouleh, Fattoush, Samboosik, and Balaleet are widely available and cost-effective, whereas sweets like Konafah, Ma’amool, Qatayif, Basbousa, and Baklava are more energy-dense and often found in modern bakeries or upscale restaurants, particularly during festive seasons. This context underscores the importance of considering accessibility, cultural preference, and ingredient sourcing when planning MNT or public health interventions.

Ingredient-based analysis confirmed that sweets mainly consist of refined carbohydrates and added sugars or fats. When consumed frequently, they may increase the risk of obesity, type 2 diabetes, and other diet-related NCDs, a trend already well documented across the region [26]. Whereas Appetizers, in contrast, provided nutrient-dense sources of plant-based protein and fiber from legumes, healthy fats from sesame paste, and vegetable-based exchanges. The heterogeneity observed in macronutrient composition across dishes reflects the diversity of preparation methods, cooking techniques, and ingredient selection. Fried items exhibited higher fat content, whereas vegetable-rich dishes offered lower carbohydrate loads with beneficial fats, highlighting how cooking methods influence energy density and nutrient quality. These insights can guide dietitians in designing culturally sensitive menus that balance macronutrient intake, portion sizes, and overall dietary quality.

The correlation analysis of the collected food composition data with values generated from food software for selected foods revealed a relatively weaker correlation for fats in appetizers. This discrepancy may be attributed to the fact that the Food Processor software data were obtained by summing the analysis of all ingredients in their raw state, while the proximate analysis was conducted on cooked foods. It is well known that cooking methods can cause variations in the macronutrient composition of the same food [27].

### 4.2. Macronutrient Findings and Clinical Implications

The nutrient comparison between international exchange standards and the developed GCC exchange list highlights the cultural differences. For starch exchanges, while the calculated mean carbohydrate (15.3 g) and protein (3.7 g) values closely match international recommendations, the mean fat content (4.1 g) is notably higher. This pattern is consistent with findings in culturally specific models like the Jordanian exchange list by Bawadi et al. [28] and the Lebanese exchange list by Hoteit et al. [29]. Traditional dishes in the Middle East often contain higher fat levels due to the use of oils, clarified butter (samn), and tahini, patterns that we also observed in our results. Similarly, fat exchanges in the GCC list match the 5 g international standard but show small amounts of carbohydrate and protein due to composite ingredients such as tahini and nuts, consistent with observations in Lebanese and Jordanian exchange lists where mixed appetizers contain non-fat nutrients. These comparisons reinforce the need for region-specific FELs rather than relying solely on international models.

When comparing our findings with existing food exchange lists of Jordan [28] and Lebanon [29] systems, the macronutrient values and portion sizes for dishes common across studies are largely consistent. This similarity is expected, as all three studies applied the standardized methodology developed by Wheeler et al., resulting in comparable exchange-unit calculations [24,25]. However, despite these methodological similarities, both the Jordanian and Lebanese exchange lists include only a limited selection of Gulf-related dishes, leaving a substantial gap in representing Qatari and broader GCC traditional foods. Our work addresses this gap by developing exchange values for culturally specific dishes that have not been previously standardized, thereby extending the international exchange-list literature into a region that has been underrepresented. By providing the first comprehensive, culturally adapted GCC exchange list, this study complements existing global models while offering clinicians and researchers a tool that aligns more closely with the dietary patterns of Gulf populations.

In alignment with our findings, Bawadi et al. [28] also reported that desserts in Jordanian cuisine mainly consist of refined carbohydrates and added sugars or fats, contributing to a higher energy density and posing potential dietary risks if consumed frequently. This trend reflects the increasing prevalence of obesity and other non-communicable diseases NCDs in the region. Hoteit et al. [29] noted that traditional Lebanese sweets required careful adaptation within exchange systems due to their sugar and fat content, while appetizers made from legumes, sesame, and vegetables provided higher nutritional value and beneficial macronutrients. Both studies also observed considerable variability across dishes, reflecting differences in cooking techniques and ingredient choices, consistent with the heterogeneity highlighted in our results. Our study further extends these insights by quantifying how methods such as frying and ingredient density can increase fat content, while dishes rich in vegetables and legumes provide opportunities for fiber intake and healthy fats.

Overall, the exchange list highlights the dual role of these traditional foods in contributing essential nutrients (legumes, seeds, and vegetables) but also exposing consumers to increased energy density and fat intake, especially when frying or sugar syrups are used. While traditional Qatari dishes are culturally and socially significant, they are predominantly energy-dense and nutrient-imbalanced, underscoring the need for moderation and portion control within dietary counseling. Importantly, the development of a culturally adapted exchange list not only facilitates accurate meal planning for diabetes and weight management but also bridges the gap between modern dietary recommendations and traditional eating practices, enabling dietitians in GCC and Qatar to deliver nutrition education that is both evidence-based and culturally relevant [14,15,30].

The findings from this study underline the clinical value of a validated food exchange list (FEL) in supporting medical nutrition therapy (MNT), empowering dietitians to personalize meal plans that improve adherence and glycemic control [14]. Multiple international and regional studies have established that culturally adapted FELs support better metabolic outcomes in patients with diabetes and cardiovascular disease [7,12]. The use of cultural food patterns and traditional meal patterns in diet-related NCDs has been shown to improve patient compliance, satisfaction, and long-term metabolic outcomes [14,15,29]. Culturally relevant food exchange lists are also adapted worldwide in countries like Ecuador (South America) [31], China [32], Lebanon [29], Jordan [33], Mali [34], Samoa [35], Spain [36], Korea [37], Greece [38], and the Philippines [39]. Some of these countries have tailored FELs to target special needs for a specific population. For example, China developed FEL with a focus on pregnant women [32], while Mali created FEL, which focused on patients with diabetes [34]. These targeted adaptations emphasize the relevance and effectiveness of FELs for addressing distinct nutritional requirements within different demographic groups. A systematic review of these FELs has demonstrated their effectiveness in managing chronic NCDs [30].

### 4.3. Comparison with International and Regional Culturally Adapted Exchange Systems

Compared with the Jordanian exchange lists [28,33], the present GCC list reports higher average carbohydrate (45–60 g vs. 30–50 g per serving) and fat content (15–30 g vs. 10–20 g), reflecting the greater prevalence of deep-fried and syrup-soaked items characteristic of Gulf cuisine (e.g., luqaimat, khanfaroosh, zalabia) and the liberal use of ghee and concentrated sugar syrups. The Lebanese system [29], although culturally proximate, records lower values (CHO 25–40 g, fat 12–25 g) owing to smaller traditional portion sizes for Levantine sweets such as baklava and knafeh. Asian and African examples reveal even clearer contrasts. The Chinese pregnancy-specific list [32] and Pakistani cultural exchange system [30] center on rice- and wheat-based sweets with more uniform carbohydrate content (≈40–50 g) but markedly lower fat (<15 g) because frying is less dominant. Similarly, the Ecuadorian [31], Malian [34], and Samoan [35] lists were developed primarily for staple-based traditional dishes rather than the highly energy-dense confectionery that predominates in the GCC region.

In contrast to the generic American Diabetes Association/American Dietetic Association exchange lists [40], which typically exclude most Arab sweets, the current GCC-specific tool permits their controlled inclusion. This approach aligns with robust evidence that culturally congruent dietary prescriptions significantly enhance long-term adherence and clinical outcomes among minority and migrant populations [41]. Table 7 presents a comparative overview of culturally adapted food exchange lists with a focus on sweets and appetizers.

### 4.4. Strengths and Limitations

One of the main strengths of this study is the reliability of the data utilized. The correlation analysis indicated a strong consistency between ESHA and local food composition tables concerning macronutrients, which boosts confidence in the generated exchange values.

Although this study provides a scientifically grounded and culturally tailored exchange list for the Qatari and wider GCC population, clinical and behavioral validation remains an essential next step. A primary limitation is the reliance on standardized recipes, which may not fully capture household-level variations in ingredients and cooking methods. While we used fixed ingredient weights, defined preparation techniques, and ESHA Food Processor software to minimize discrepancies, natural differences in food preparation and ingredient sourcing may still introduce minor variability in macronutrient content. These differences are not expected to meaningfully alter exchange classifications; however, users should consult the listed recipe ingredients when modifying dishes for dietary planning.

Evidence from culturally adapted exchange-list research supports this standardized approach. In Jordan, Bawadi et al. (2008) demonstrated that using fixed ingredient weights, controlled cooking methods, and defined portion sizes keeps nutrient variability within acceptable exchange-category limits [33]. Similarly, Hoteit et al. (2021) showed that standardized Lebanese and Eastern Mediterranean recipes yield consistent macronutrient profiles despite regional preparation differences [29]. Spanish work by Marques-Lopes et al. (2018) further confirms that foods grouped into exchanges remain nutritionally interchangeable when recipes are standardized [36]. Likewise, the ADA/AND system (2019) bases all exchanges on consistent carbohydrate, protein, fat, and calorie values, reinforcing the importance of recipe standardization for reliable exchange-list classification [40].

### 4.5. Practical, Policy, and Future Work Implications

The developed food exchange list holds substantial potential for integration into clinical practice, public health policy, and digital nutrition platforms. Its application in meal planning, nutrition education, and patient counseling can enhance the accuracy and cultural relevance of dietary guidance across the region. Incorporating this system into national nutrition guidelines would strengthen sustainable dietary management efforts and promote healthier eating behaviors at the population level.

Moreover, the exchange list provides a foundational framework for embedding culturally appropriate dietary planning within national food policies and digital dietary assessment tools. Its adoption in healthcare settings would enable dietitians to deliver personalized, evidence-based counseling that aligns with regional food traditions, improving adherence and long-term health outcomes.

Beyond clinical applications, it provides a framework for public health initiatives, diabetes prevention programs [31,41], nutrition education [31], and policy formulation [42], supporting sustainable, culturally appropriate dietary management. To operationalize these policy recommendations, the exchange list should first be piloted within Qatar University Health Clinics and selected Primary Health Care Corporation (PHCC) centers. Training for dietitians would cover standardized portion size estimation, exchange calculation, culturally adapted menu planning, and case-based counseling scenarios. Collaboration with the Ministry of Public Health (MoPH), the Ministry of Education and Higher Education, and PHCC will facilitate integration into national nutrition guidelines, school curricula, and community health programs. Structured training workshops and a monitoring and feedback system for clinicians and patients will support consistent application, continuous refinement, and cultural appropriateness of the tool. The exchange list can also be incorporated into university nutrition programs and school health curricula to strengthen nutrition literacy and prepare future health professionals in culturally relevant dietary-planning tools. Following successful pilots, a coordinated national rollout, developed with public health agencies, would enable broader adoption across dietetic clinics and population-wide nutrition campaigns. For digital applications, the system can be embedded into mobile health apps, web platforms, and electronic medical records to support individualized meal planning and real-time nutrient tracking.

Looking ahead, future work should assess the usability, validity, and clinical effectiveness of the exchange list across diverse populations and settings to guide its integration into national strategies, dietetic education, and digital health systems, ensuring scalability and sustained impact.

## 5. Conclusions

This study provides the first GCC-specific, culturally adapted macronutrient-based exchange list for traditional Qatari and Gulf sweets and appetizers, addressing a major gap in regional dietary planning resources. The nutrient comparison demonstrated strong correlations between laboratory-analyzed regional food composition tables and ESHA-derived values (r = 0.78–0.92, *p* < 0.05), confirming the reliability of the standardized recipes and macronutrient estimates. The analysis also revealed notable variability in fat and carbohydrate content among dishes, particularly in fried sweets and sesame-based appetizers, highlighting the importance of culturally tailored portion control within nutrition counseling. By aligning all portion sizes with both Wheeler et al. exchange standards and the Qatar Dietary Guidelines, this work establishes a scientifically rigorous framework for translating traditional dishes into standardized exchange units that dietitians can use in clinical and community settings [25].

Although the exchange list is based on validated nutrient data and standardised preparation procedures, minor variability may persist due to differences in traditional cooking methods and ingredient sourcing. Additionally, the tool has not yet undergone clinical or behavioral validation, which limits its immediate applicability in intervention-based settings.

Future work should therefore focus on evaluating usability, metabolic outcomes, and patient adherence through structured pilot studies in Qatari and GCC populations. Further efforts are also warranted to integrate the exchange list into digital dietary assessment tools, apply it in nutritional epidemiology, and incorporate it into national nutrition guidelines and public health initiatives to support culturally relevant, evidence-based dietary management across the region.

## Figures and Tables

**Figure 1 nutrients-18-00117-f001:**
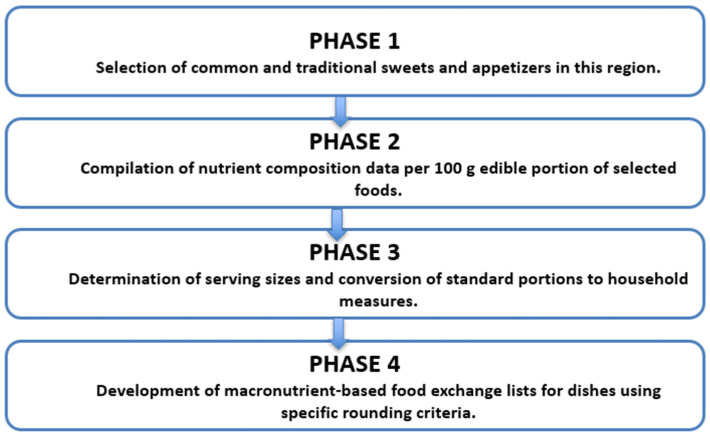
Conceptual framework followed in developing the food exchange list, illustrating the four sequential phases applied to all analyzed dishes (*n* = 34).

**Figure 2 nutrients-18-00117-f002:**
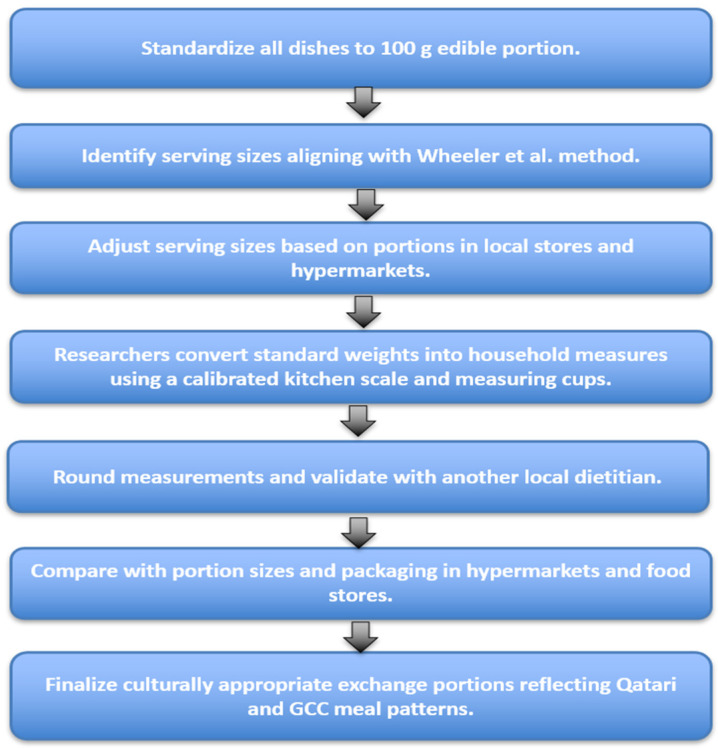
Methodological steps used to define exchange portions for all analyzed dishes (*n* = 34) [24].

**Table 1 nutrients-18-00117-t001:** Sources of food composition data for selected traditional Qatari/GCC dishes, and preparation type.

Food Composition Data Source	No. of Food Items	Selected Food Items	Preparation Type
Study conducted by El Obeid [18]	2	**Sweets:** Sago (Sagau), Barinoish	Domestically prepared
Study conducted by Nagdy et al. [19]	2	**Sweets:** Aigalee, Assedah	Domestically prepared under the supervision of a qualified dietitian.
Food composition table of Bahrain [20]	9	**Sweets:** Balleleet, Mahamer, Khabeese, Khanfarooshe, Hesso Elbah, Luqemat, Betheeth Muhalbiya	Domestically prepared
Jordanian Food composition table [21]	21	**Sweets:** Konafah na’ema bil jibn, Qatayif bil jibn Maqli, Qatayif bil jooz Maqli, Qatayif bil jibn Mashwi, Qatayif bil jooz Mashwi, Ma’amool ajwa bilsameed, Ma’amool ajwa bil taheen, Mamool bil jooz, Mamool bil fustok **Appetizers:** Motabbal bathijan bil tahina, Hummus bil tahina, Foul modammas, Sambosik bil sabanikh, Falafel, Motabbal bathinjan bil khodar, Manaqeesh za’ atar, Tabbouleh, Fatoosh, Mosabbaha, Baqdonsieh, Za atar mix	Commercially purchased

**Table 3 nutrients-18-00117-t003:** Macronutrient correlation analysis.

Dish Group	Major Ingredients	Carbohydrate (g/100 g)	Protein (g/100 g)	Fat (g/100 g)	*p*-Value
Sweets *n* = 22	Grain/Flour, Sweetener, Dairy/egg, Fats & oils, Nuts	0.78	0.72	0.80	<0.05
Appetizers *n* = 12	Grains/Bread, Legumes/Pulses, Vegetables	0.90	0.92	0.53	<0.05

Note: *p* < 0.05 is considered statistically significant.

**Table 4 nutrients-18-00117-t004:** Traditional and common GCC and Qatari sweet exchange list based on standardized serving portions, showing equivalent macronutrient values.

Dish Names	Description	Serving Weight	Serving Size	Carbohydrates (g)	Protein (g)	Fat (g)	Exchange/ Serving
Balleleet	Sweetened vermicelli flavored with cardamom and saffron	100 g	1 small to medium piece	30.4	5.5	6	1 Starch, 1 Other carbohydrate, 1 Fat
Mahamer	Fried Sweet Rice (sucrose & pure corn oil)	100 g	½ cup	49	2.9	4	1.5 Starch, 1.5 Other carbohydrates, 1 Fat
Barinoish	Sweet made from rice	100 g	½ cup	36.8	2	0	2 Starch, 0.5 Other Carbohydrates
Sago (Sagau)	Sago Pudding	100 g	½ cup	58	1.9	0.1	2 Starch, 2 Other Carbohydrates
Khabeese	Flavored Semolina cake with sugar syrup and cardamom	100 g	2 small pieces	58.6	3.5	6.3	2 Starch, 2 Other Carbohydrates, 1 Fat
Khanfarooshe	Dessert made from ground rice and egg	100 g	2 pieces	50.1	6.2	10.1	1.5 Starch, 1.5 Other carbohydrates, 1 MFM, 1 Fat
Hesso	Egg and watercress sweet	100 g	½ cup	15	2.9	7.1	1 Starch, 1 Fat
Elbah	Egg with milk pudding	100 g	½ cup	18	4.3	3.3	1 Other Carbohydrate, 0.5 Low-fat milk
Aigalee	Baked Cardamom cake with egg	100 g	1 medium slice	19.1	8.4	35.8	1 starch, 1 vegetable, 1 MFM, 6 fat
Asseda (assedah)	Pudding-like dessert with wheat flour	100 g	½ cup	17.1	3.1	29	1 Starch, 5 Fat
Luqemat/ Awameh	Sweet dumplings soaked in sugar syrup	40 g	2 small pieces	28.1	1.2	2.6	1 Starch, 1 Other Carbohydrate, 0.5 Fat
Muhalbiya	Rice Pudding	100 g	½ cup	17.6	3	3	1 Starch, 0.5 Fat
Betheeth	Dates with Flour	30 g	1 piece	21.3	0.7	2	1 Starch, 0.5 Fat
Konafah na’ema bil jibn	Layers of pastry and cheese filled with sugar syrup and cheese	100 g	Medium-sized piece (piece)	36	10.7	16.4	1 Starch, 1 Other carbohydrate, 1 MFM, 0.5 Whole-fat milk
Qatayif bil jibn Maqli	Deep-fried pancake-like dough stuffed with cheese and dipped in sugar syrup	40 g	1 piece	16.6	4	4	1 Starch, 1 Fat
Qatayif bil jooz Maqli	Deep-fried pancake-like dough stuffed with walnuts and dipped in sugar syrup	45 g	1 piece	26.5	2.2	4.5	1 Starch, 0.5 Other carbohydrate, 1 Fat
Qatayif bil jibn Mashwi	Grilled pancake-like dough stuffed with cheese and dipped in sugar syrup	50 g	1 piece	12.3	3.1	1.7	1 Starch
Qatayif bil jooz Mashwi	Grilled pancake-like dough stuffed with walnuts and dipped in sugar syrup	50 g	1 piece	14	3.7	8	1 Starch, 1.5 Fat
Ma’amool ajwa bilsameed	semolina cookies stuffed with dates	30 g	1 piece	17.9	2.7	5.8	1 Starch, 1 Fat
Ma’amool ajwa bil taheen	Semolina cookies stuffed with dates and sesame paste	30 g	1 piece	18.6	2.7	5.3	1 Starch, 1 Fat
Mamool bil jooz	Semolina cookie filled with walnuts	30 g	1 piece	13.3	2.5	7.3	1 Starch, 1 Fat
Mamool bil fustok	A cookie filled with pistachio	35 g	1 piece	6	2	2.6	0.5 Starch. 0.5 Fat

Abbreviations: MFM: medium fat meat. Note: The term “Starch” refers to the “Carbohydrate Exchange” food group, where one exchange is approximately 15 g of carbohydrate and 80 kcal. All nutrient values provided correspond to the serving weight indicated in the table.

**Table 5 nutrients-18-00117-t005:** Traditional and common GCC and Qatari appetizers exchange list based on standardized serving portions with equivalent macronutrient values.

Dish Names	Description	Serving Weight	Serving Size	Carbohydrates (g)	Protein (g)	Fat (g)	Exchange/ Serving
Motabbal bathijan bil tahina	Roasted eggplant and sesame blend as a dip	100 g	1/2 cup	6.9	3.5	10.9	1 Vegetable, 2 Fat
Hummus bil tahina	Boiled chickpeas and sesame paste blend as a dip	100 g	1/2 cup	14	7.5	10.6	1 Starch, 1 LM, 1.5 Fat
Foul modammas	Fava bean dip	100 g	1/2 cup	14.7	7.6	4.9	1 Starch, 1 LM, 1 Fat
Sambosik bil sabanikh	Fried spinach-stuffed pastry	70 g	1 medium-sized piece	24.8	4	8.8	1 Starch 1.5 Vegetable, 1.5 Fat
Falafel	A paste of broad beans and/or chickpeas that is deep-fried in oil	100 g	4 pieces	27.1	11.1	12.6	1.5 Starch, 1 LM, 2 Fat
Motabbal bathinjan bil khodaar	Roasted eggplant and vegetables blend as a dip	100 g	1/2 cup	5.6	1.3	5.2	1 Vegetable, 1 Fat
Manaqeesh za’ atar	Bread with dried thyme and sesame pie	50 g	½ piece	23.4	5.1	12.5	1 Starch, 2 Fat
Tabbouleh	Salad of parsley, tomatoes, onion, lemon juice, olive oil, and bulghur	100 g	3/4 cup	8.6	2.9	10.9	1 Vegetable. 2 Fat
Fatoosh	Vegetable salad with fried bread	100 g	1 cup	7.6	2.1	5.9	1 Vegetable, 1 Fat
Mosabbaha	Boiled chickpeas, yogurt, and tahini blend as a dip	100 g	1/2 cup	15.8	7.4	9.5	1 Starch, 1 LM, 1.5 Fat
Baqdonsieh	Dip with sesame paste	100 g	1/2 cup	5.5	7.6	19.6	0.5 Vegetable, 1 LM, 3.5 Fats
Za atar mix	Arabic spice blend	20 g	2 tbsp	4.5	3.7	6.5	1 Vegetable, 1 Fat

Abbreviations: LM: lean meatNote: The term “Starch” refers to the “Carbohydrate Exchange” food group, where one exchange is approximately 15 g of carbohydrate and 80 kcal. All nutrient values provided correspond to the serving weight indicated in the table.

**Table 6 nutrients-18-00117-t006:** Comparison of the mean calculated GCC exchange profile with the standard international value.

Exchange Group	Standard International Value (Approximate)	Mean Calculated Nutrient Profile in GCC List (g/Exchange)	Key Finding
Starch (1 Carbohydrate serving)	15 g Carbohydrate 3 g Pro 0–1 g Fat	15.3 ± 4.1 g Carbohydrate 3.7 ± 1.4 g Pro 4.1 ± 3.2 g Fat	CHO and protein are consistent. Fat content is higher, reflecting the use of clarified butter (samn) and oils in traditional preparation.
Fat (1 Fat serving)	5 g Fat 0 g Carbohydrate 0 g Protein	5.1 ± 1.4 g Fat 3.1 ± 1.5 g Carbohydrate 1.2 ± 0.9 g Protein	Fat content is consistent. Small amounts of CHO/Pro are noted, attributed to composite ingredients (e.g., tahini and nuts) in appetizers.

**Table 7 nutrients-18-00117-t007:** Comparative overview of culturally adapted food exchange lists with a focus on sweets and appetizers.

Region/Country	Year	No. of Sweet/Appetizer Items	Mean CHO per Serving (g)	Mean Fat per Serving (g)	Key Cultural Feature Addressed	Reference
**Jordan**	2009	40 (20 appetizers + 20 desserts)	30–50	10–20	Levantine appetizers & wheat-based desserts	[28]
**Lebanon**	2021	30+ (dishes + Arabic sweets)	25–40	12–25	Baklava/knāfeh sweets; smaller Levantine portions	[29]
**Pakistan**	2017	Limited (milk/grain sweets)	~45	<15	Rice/wheat-based, low-fat confections	[30]
**China (pregnancy)**	2023	Included (sweet snacks)	40–50	10–15	Rice/flour-based; pregnancy-adapted	[32]
**Ecuador**	2023	General (sweets/staples incl.)	35–45	<10	Plantain/tuber-based; low-fat staples	[31]
**Mali (diabetes)**	2009	Limited (sugar-added foods)	15–25	<5	Local staples; minimal sweets for T2D control	[34]
**Samoa**	1994	Culturally accepted (dishes/sweets)	35–45	10–15	Coconut-rich; high sodium/fat highlights	[35]
**GCC/** **Qatar**	2025	34	45–60	15–35	Syrup/ghee-fried sweets dominant; legume/veg appetizers	This study

## Data Availability

The data presented in this study are available on request from the corresponding author due to ongoing publication processes.

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
