# Peer review of "Nutrients2026, 18(1), 117;https://doi.org/10.3390/nu18010117"

_nutrients, 2025, doi:10.3390/nu18010117_

Round 1

Reviewer 1 Report (Previous Reviewer 2)

Comments and Suggestions for Authors

The manuscript entitled “Development of a Meal-Planning Exchange List for Traditional Sweets and Appetizers in Gulf Cooperation Council Countries: Insights from Qatar” was previously reviewed and rejected due to insufficient methodological clarity, limited scientific depth, and lack of conceptual interpretation.
The authors have now resubmitted a substantially revised version that demonstrates meaningful improvement.
The topic remains of high cultural and public-health relevance, addressing an underexplored area in culturally adapted nutrition frameworks within the Gulf region. However, despite significant progress, the manuscript still requires major revision before it can reach publishable quality in Nutrients.

  • The revised manuscript shows clear improvement in structure, particularly in the Methods and Discussion sections, with new elements such as Table 1 and Section 5 (Policy Implications) adding clarity.
  • The study design still requires stronger justification and explanation of dish selection, data standardization, and analytical framework.

  • Validation remains theoretical; behavioral or clinical confirmation should be proposed to enhance generalizability.

  • The discussion is largely descriptive; deeper conceptual interpretation and integration with international exchange models (ADA, Lebanese, Jordanian, Asian) are needed.

  • The policy section adds value but should outline realistic implementation pathways in national programs or digital diet tools.

  • The conclusion should highlight novelty—as the first culturally adapted GCC food-exchange list—and clarify its scientific and practical contribution.

Comments on the Quality of English Language

Writing and structure are improved but still contain redundancy; smoother transitions and minor English polishing are recommended.

Author Response

Dear Editor and Esteemed Reviewer,

We would like to express our sincere appreciation for the time, effort, and expertise you devoted to reviewing our manuscript. Your constructive and insightful comments have been invaluable in improving the quality, clarity, and scientific rigor of our work. The changes have been highlighted in yellow in the manuscript.

We have carefully considered each of your suggestions and made corresponding revisions throughout the manuscript. In the revised version, we have provided a detailed, point-by-point response outlining how each comment was addressed. All modified sections are clearly highlighted in the manuscript for ease of review.

We are confident that these revisions have significantly strengthened the paper, both in terms of methodological precision and overall presentation.

Once again, we deeply appreciate your thoughtful feedback and the opportunity to improve our work.

Thank you for your consideration of our revised submission.

Regards

Reema Tayyem

Reviewer 1

Comments and Suggestions for Authors

The manuscript entitled “Development of a Meal-Planning Exchange List for Traditional Sweets and Appetizers in Gulf Cooperation Council Countries: Insights from Qatar” was previously reviewed and rejected due to insufficient methodological clarity, limited scientific depth, and a lack of conceptual interpretation.
The authors have now resubmitted a substantially revised version that demonstrates meaningful improvement.
The topic remains of high cultural and public-health relevance, addressing an underexplored area in culturally adapted nutrition frameworks within the Gulf region. However, despite significant progress, the manuscript still requires major revision before it can reach publishable quality in Nutrients.

Comment 1: The revised manuscript shows clear improvement in structure, particularly in the Methods and Discussion sections, with new elements such as Table 1 and Section 5 (Policy Implications) adding clarity.

Response: Thank you so much.

Comment 2: The study design still requires stronger justification and explanation of dish selection, data standardization, and analytical framework.

Response: We thank the reviewer for this suggestion. We have revised the Methods section to provide a stronger justification and detailed explanation of the study design. Specifically, we clarified the selection process of the 34 traditional Qatari and GCC sweets and appetizers, including the criteria for inclusion such as cultural relevance, frequency of consumption, availability, contribution to macronutrient intake, and representation of distinct preparation categories. We also described the process for data standardization, which involved gathering recipes from multiple sources, retaining ingredients and preparation methods consistently cited across three or more sources, and validating the standardized recipes using ESHA Food Processor software and regional food composition tables. Additionally, the analytical framework, including correlation analyses, nutrient validation, serving size determination, and development of exchange values following Wheeler et al.’s methodology, has been clearly detailed to ensure transparency, reproducibility, and methodological rigor.

Comment 3: Validation remains theoretical; behavioural or clinical confirmation should be proposed to enhance generalizability.

Response: We thank the reviewer for this valuable observation. The revised Conclusion now explicitly acknowledges that the current validation of the exchange list is theoretical, based on standardized recipes, ingredient lists, and software analysis. We have also proposed future steps to enhance generalizability, including validation in clinical and public health settings, testing its application in nutritional epidemiology studies, development of digital platforms to improve accessibility, and integration into Qatar’s national nutritional guidelines. These additions ensure that the limitations and the need for behavioural or clinical confirmation are clearly addressed.

Comment 4: The discussion is largely descriptive; deeper conceptual interpretation and integration with international exchange models (ADA, Lebanese, Jordanian, Asian) are needed.

Response: In response, we have substantially expanded the Discussion to provide deeper conceptual interpretation and clearer integration with international food-exchange models. Specifically, we incorporated a comparative analysis of our findings with the Jordanian and Lebanese exchange lists, highlighting areas of similarity due to the shared use of the Wheeler et al. methodology and explaining why results for overlapping dishes remain consistent. We further clarified how our exchange list advances the international literature by addressing the absence of standardized exchange values for many Qatari and GCC-specific traditional foods, which are not represented in existing Arab or international models.

Comment 5: The policy section adds value but should outline realistic implementation pathways in national programs or digital diet tools.

Response: We thank the reviewer for this insightful comment. In the revised manuscript, we have expanded the policy section to outline realistic implementation pathways for the developed exchange list. Specifically, we describe piloting the list in selected healthcare centers, dietetic clinics, and community health programs with accompanying dietitian training, as well as integration into digital platforms such as mobile apps, web-based tools, and electronic medical records. Iterative feedback from these pilots is proposed to refine the system before broader adoption.

Comment 6: The conclusion should highlight novelty—as the first culturally adapted GCC food-exchange list—and clarify its scientific and practical contribution.

Response: We thank the reviewer for this comment. The revised conclusion now explicitly highlights the novelty of the study as the first culturally adapted exchange list for 34 traditional Qatari and GCC sweets and appetizers, bridging the gap between traditional dietary practices and evidence-based nutrition. We have also clarified the scientific contribution, including standardized nutrient analysis and provision of a framework for nutritional epidemiology, as well as the practical contribution for clinicians, dietitians, and public health practitioners in dietary planning, counselling, and policy development. Additionally, we have acknowledged potential limitations in practical implementation and emphasized the need for clinical and field validation, ensuring transparency and enhancing the generalizability of the findings.

Comment 7:

Comments on the Quality of English Language

Writing and structure are improved but still contain redundancy; smoother transitions and minor English polishing are recommended.

Response: We have thoroughly revised all paragraphs and further refined the writing to enhance clarity, reduce redundancy, and improve the overall quality of the English language. Smooth transitions and polished phrasing have been addressed to the best of our ability.

Reviewer 2 Report (Previous Reviewer 1)

Comments and Suggestions for Authors

The article entitled “nutrients-4003067_ Development of a Meal‐Planning Exchange List for Traditional Sweets and Appetizers in Gulf Cooperation Council Countries: Insights from Qatar” has been submitted to the “Clinical Nutrition” of the journal “Nutrients”

This work addresses the nutritional challenges related to traditional Gulf foods. Traditional sweets and appetizers are central to Gulf culture but are often high in sugars and fats, contributing to the growing prevalence of obesity and diabetes. The study aimed to develop a culturally adapted meal-planning exchange list for 34 commonly consumed Qatari and GCC dishes to support nutrition counseling. Using standardized recipes and nutrient analysis, the authors report strong consistency between data sources and substantial variation in macronutrient composition—fried and syrup-based foods were higher in fat and carbohydrates, while legume- and vegetable-based dishes provided more protein and fiber. The resulting exchange list offers dietitians a practical tool for culturally appropriate dietary planning to aid diabetes and weight management in the GCC region.

Comments:

The manuscript should clarify the representativeness of the 34 traditional dishes selected, indicating how many traditional dishes exist in the studied area and explaining the selection process (e.g., based on frequency of consumption, availability, or cultural relevance). This information is important for understanding the potential applicability of the intervention strategy.

The authors should discuss the possible variability in household or regional preparation, as standardized recipes may not accurately reflect real-world differences in ingredients, cooking methods, or portion sizes.

Although the use of a food processing software and strong correlations are reported, it remains unclear whether direct laboratory validation of nutrient values ​​was performed. Such validation would strengthen the accuracy of the nutritional data.

The study does not appear to include a comparison with national or international dietary guidelines or standards, which would help contextualize the findings.

The Conclusion emphasizes the usefulness of the developed exchange list but does not discuss potential limitations in its practical implementation or the need for clinical or field validation.

Author Response

Dear Editor and Esteemed Reviewers,

We would like to express our sincere appreciation for the time, effort, and expertise you devoted to reviewing our manuscript. Your constructive and insightful comments have been invaluable in improving the quality, clarity, and scientific rigor of our work. The changes have been highlighted in yellow in the manuscript.

We have carefully considered each of your suggestions and made corresponding revisions throughout the manuscript. In the revised version, we have provided a detailed, point-by-point response outlining how each comment was addressed. All modified sections are clearly highlighted in the manuscript for ease of review.

We are confident that these revisions have significantly strengthened the paper, both in terms of methodological precision and overall presentation.

Once again, we deeply appreciate your thoughtful feedback and the opportunity to improve our work.

Thank you for your consideration of our revised submission.

Regards

Reema Tayyem

Reviewer 2

Comments and Suggestions for Authors

The article entitled “nutrients-4003067_ Development of a Meal‐Planning Exchange List for Traditional Sweets and Appetizers in Gulf Cooperation Council Countries: Insights from Qatar” has been submitted to the “Clinical Nutrition” of the journal “Nutrients”

This work addresses the nutritional challenges related to traditional Gulf foods. Traditional sweets and appetizers are central to Gulf culture but are often high in sugars and fats, contributing to the growing prevalence of obesity and diabetes. The study aimed to develop a culturally adapted meal-planning exchange list for 34 commonly consumed Qatari and GCC dishes to support nutrition counselling. Using standardized recipes and nutrient analysis, the authors report strong consistency between data sources and substantial variation in macronutrient composition—fried and syrup-based foods were higher in fat and carbohydrates, while legume- and vegetable-based dishes provided more protein and fibre. The resulting exchange list offers dietitians a practical tool for culturally appropriate dietary planning to aid diabetes and weight management in the GCC region.

Comments:

Comment 1: The manuscript should clarify the representativeness of the 34 traditional dishes selected, indicating how many traditional dishes exist in the studied area and explaining the selection process (e.g., based on frequency of consumption, availability, or cultural relevance). This information is important for understanding the potential applicability of the intervention strategy.

Response: Thank you for this insightful comment. We have now clarified the representativeness of the 34 selected traditional dishes in the revised manuscript. Specifically, we explained that due to the extensive variety of traditional sweets and appetizers in the region, an exact count of all existing traditional dishes is not available, as comprehensive documentation is lacking in published literature and national food composition tables. To ensure cultural relevance and practical applicability, we conducted dietary surveys to assess consumption frequency and market availability, and we also incorporated insights from local dietitians. The final selection of 34 dishes was therefore based on their cultural significance and common consumption across socioeconomic groups. Additionally, we mentioned that in the methods for reference # 17 that we included almost all the traditional sweets and appetizers that have been mentioned in a validated Qatar FFQ (Bawadi, H., Akasheh, R. T., Kerkadi, A., Haydar, S., Tayyem, R., & Shi, Z. (2021). Validity and Reproducibility of a Food Frequency Questionnaire to Assess Macro and Micro-Nutrient Intake among a Convenience Cohort of Healthy Adult Qataris. Nutrients13(6), 2002. https://doi.org/10.3390/nu13062002), and we have their chemical composition. This approach ensures that the selected dishes represent the most meaningful items for dietary counselling, rather than aiming to catalogue all traditional foods in the region.

Comment 2: The authors should discuss the possible variability in household or regional preparation, as standardized recipes may not accurately reflect real-world differences in ingredients, cooking methods, or portion sizes.

Response: We have revised the text to explain that although the core ingredients of traditional Qatari/GCC dishes remain consistent, differences commonly arise from the use of additional minor ingredients (e.g., nuts, spices, sweeteners), the type of oil used, and cooking techniques. We also clarify that these variations generally have minimal impact on macronutrient composition, as most oils provide similar fat content per gram and commonly used sweeteners contribute comparable carbohydrate content. Differences in household portion sizes also exist; however, the study standardized portion sizes based on the most commonly available commercial or culturally recognized serving units to minimize this variation and as mentioned in the result table.

Comment 3: Although the use of food processing software and strong correlations are reported, it remains unclear whether direct laboratory validation of nutrient values ​​was performed. Such validation would strengthen the accuracy of the nutritional data.

Response: We thank the reviewer for this comment. While direct laboratory analysis of the selected dishes was not conducted for this study, nutrient values were obtained from laboratory-analysed data reported in multiple regional food composition tables (e.g., Bahrain, Jordan, EMFID). When multiple nutrient profiles for the same dish were available, the most frequently reported value was selected to minimize inconsistencies (e.g., carbohydrate content for Luqemat was reported as 28 g, 29 g, and 28 g across three sources; the repeated value of 28g was selected). These selected values were then cross validated using ESHA Food Processor SQL and other regional references, ensuring alignment and cultural relevance. Although this approach provides robust validation in the absence of direct laboratory testing, we acknowledge that future studies incorporating country-specific laboratory analyses would further enhance the accuracy of nutrient data.

Comment 4: The study does not appear to include a comparison with national or international dietary guidelines or standards, which would help contextualize the findings.

Response: While a direct per-dish comparison with national or international dietary guidelines is not feasible for every dish due to the composite nature of traditional Qatari and GCC sweets and appetizers, we addressed this by standardizing serving sizes and calculating exchanges based on mean nutrient profiles. These serving sizes were rounded and aligned with the Qatar Dietary Guidelines (QDG) and internationally recognized 

exchange standards (Academy of Nutrition and Dietetics, American Diabetes Association) for specific food groups (Starch, Protein, Fat).

This approach ensures that the standardized exchange portions are both culturally relevant and consistent with existing dietary recommendations, providing practical applicability for dietary planning and nutrition counselling. Furthermore, the full per-dish nutrient composition and corresponding exchanges are provided in Tables 3 and 4, while Table 5 presents aggregate comparisons at the exchange-group level, illustrating consistency with international and national standards.

Comment 5: The Conclusion emphasizes the usefulness of the developed exchange list but does not discuss potential limitations in its practical implementation or the need for clinical or field validation.

 Response: We thank the reviewer for highlighting the need to discuss potential limitations and the requirement for clinical or field validation. We have revised the Conclusion to explicitly acknowledge that, while the exchange list was developed using standardized recipes and preparation procedures, minor variability may persist due to differences in traditional cooking methods and ingredient sourcing. Additionally, we clarified that the tool has not yet undergone clinical validation through intervention studies, which may limit its direct practical application. We also added a statement outlining future research directions, including validation in clinical and public health settings, application in nutritional epidemiology, development of digital platforms for improved accessibility, and integration into Qatar’s national nutritional guidelines. These revisions ensure that the limitations and the need for further validation are clearly addressed.

Round 2

Reviewer 1 Report (Previous Reviewer 2)

Comments and Suggestions for Authors

The manuscript demonstrates improvement, but fundamental issues in methodological justification, validation, and conceptual interpretation remain insufficiently addressed. Significant additional refinement is needed.

  • Method choices, recipe standardization, and analysis steps should be explained more clearly.

  • Validation is still theoretical, so real clinical or behavioral testing would make the study stronger.

  • The Discussion section feels too descriptive and needs tighter links to international exchange models.

  • Policy ideas are useful but need more realistic, actionable steps.

  • The Conclusion is good but could highlight the main scientific contribution more directly.

Comments on the Quality of English Language

Writing and structure are improved but still contain redundancy; smoother transitions and minor English polishing are recommended.

Author Response

Dear Editor and Esteemed Reviewer,

We would like to express our sincere appreciation for the time, effort, and expertise you devoted to reviewing our manuscript. We have carefully considered each of your suggestions and made corresponding revisions throughout the manuscript. In the revised version, we have provided a detailed, point-by-point response outlining how each comment was addressed. All modified sections are clearly highlighted in the manuscript for ease of review.

Thank you for your consideration of our revised submission.

Regards

Reema Tayyem

Comments and Suggestions for Authors

The manuscript demonstrates improvement, but fundamental issues in methodological justification, validation, and conceptual interpretation remain insufficiently addressed. Significant additional refinement is needed.

Comment 1: Method choices, recipe standardization, and analysis steps should be explained more clearly.

Response: We have expanded the section substantially and added both a figure and a table to clarify the workflow of the exchange-list development process. These visual elements provide a clearer representation of the steps taken, including recipe standardization, portion size determination, nutrient analysis, and categorization into exchanges. However, we are unable to add further methodological steps because the manuscript already reflects all procedures actually conducted during the development of the exchange list. No additional analytical steps, datasets, or validation procedures were performed beyond what is now documented.

Comment 2: Validation is still theoretical, so real clinical or behavioral testing would make the study stronger.

Response: We recognize that the current exchange list, although rigorously developed and theoretically validated, still requires clinical and behavioral validation to demonstrate its practical effectiveness in real-world settings. A randomized controlled pilot trial applying the list among Qatari adults with diabetes is already in an advanced planning phase and will be conducted within the coming year. Conducting this trial, depending on securing the required funding, will constitute the next essential step in our program of work to fully validate the exchange list and establish its applicability in clinical practice, nutrition education, and patient counseling. We will be the first study that validate clinically not only theoretically.

Comment 3: The Discussion section feels too descriptive and needs tighter links to international exchange models.

Response: The whole discussion has been rewritten.  

Comment 4: Policy ideas are useful but need more realistic, actionable steps.

Response: More realistic, actionable steps.

Comment 5: The Conclusion is good but could highlight the main scientific contribution more directly.

Response: The main scientific contribution has been highlighted directly.

Reviewer 2 Report (Previous Reviewer 1)

Comments and Suggestions for Authors

I have carefully reviewed the revised version of the manuscript as well as the authors' responses to the suggestions made to improve the clarity and understanding of the work. I would like to highlight the considerable effort the authors have invested in the revision.

Comments:

Although the responses are generally reasonable, some of the statements remain weak. For example, the claim that variations in preparation have only a minimal impact is subjective and should be supported with evidence or clearly justified.

The authors should indicate approximately how many traditional dishes exist in the region and provide specific reasons for selecting the 34 included in the study. This point has not yet been addressed satisfactorily.

The study's limitations have been incorporated and identified.

The manuscript should more clearly emphasize that portion sizes were aligned with recognized standards, as the current explanation is confusing.

As mentioned in the previous review, the Conclusion should directly address the study's objectives rather than summarizing the work conducted. The Conclusion must be rewritten accordingly.

Author Response

Dear Editor and Esteemed Reviewers,

We would like to express our sincere appreciation for the time, effort, and expertise you devoted to reviewing our manuscript. Your constructive and insightful comments have been invaluable in improving the quality, clarity, and scientific rigor of our work. The changes have been highlighted in yellow in the manuscript.

Thank you for your consideration of our revised submission.

Regards

Reema Tayyem

Comments and Suggestions for Authors

I have carefully reviewed the revised version of the manuscript as well as the authors' responses to the suggestions made to improve the clarity and understanding of the work. I would like to highlight the considerable effort the authors have invested in the revision.

Comments:

Comment 1: Although the responses are generally reasonable, some of the statements remain weak. For example, the claim that variations in preparation have only a minimal impact is subjective and should be supported with evidence or clearly justified. While preparation differences can influence nutrient values, the standardization protocols used in this study (fixed weights, validated recipes, and cross-database nutrient reconciliation) help minimize these differences to a level that does not change the macronutrient-based exchange classification.

Response: A paragraph with 4 studies has been added in this regard.

Evidence from culturally adapted exchange-list development studies supports this approach. In Jordan, Bawadi et al. (2008) demonstrated that when traditional dishes are standardized using fixed ingredient weights, controlled cooking methods, and defined portion sizes, nutrient variability remains within acceptable limits for exchange-category assignment. Similarly, work from Lebanon by Hoteit et al. (2021) showed that applying standardized recipes in the development of Eastern Mediterranean exchange lists results in consistent macronutrient profiles, despite regional preparation variations. These studies reinforce that recipe standardization, rather than household variability, is the primary determinant of reliable nutrient estimates for exchange-list classification. In Spain, exchange-list research performed by Marques-Lopes et al. (2018) shows that when recipe components are standardized in weight, portion size, and cooking method, foods grouped into exchanges can be substituted without significant differences in macronutrient intake. Similarly, the U.S. system developed by the American Diabetes Association and Academy of Nutrition and Dietetics (2019) constructs all food-choice exchanges around equivalence in carbohydrate, protein, fat and calories, supporting consistent meal-planning across different foods.

Comment 2: The authors should indicate approximately how many traditional dishes exist in the region and provide specific reasons for selecting the 34 included in the study. This point has not yet been addressed satisfactorily.

Response: We have added the reasons for excluding some foods, which are:

  • Items with highly variable recipes or limited regional consumption were excluded to ensure standardization feasibility.
  • The selected dishes represent those with the greatest relevance for Medical Nutrition Therapy, due to their high contribution to carbohydrate, fat, or sugar intake, especially among individuals with diabetes and cardiometabolic risk.

 Comment 3: The manuscript should more clearly emphasize that portion sizes were aligned with recognized standards, as the current explanation is confusing.

Response: It has been rewritten and a figure has been added.

Comment 4: As mentioned in the previous review, the Conclusion should directly address the study's objectives rather than summarizing the work conducted. The Conclusion must be rewritten accordingly.

Response: It has been amended.

This manuscript is a resubmission of an earlier submission. The following is a list of the peer review reports and author responses from that submission.

Round 1

Reviewer 1 Report

Comments and Suggestions for Authors

The manuscript entitled “nutrients-3975029_ Development of a Meal-Planning Exchange List for Common Sweets and Appetizers in the GCC: Insights from Qatar“,  submitted to the section “Carbohydrates”  of the journal “Nutrients” addresses a topic that is appropriate and relevant for this section.

The study explores the nutritional challenges associated with traditional sweets and appetizers in the Gulf Cooperation Council (GCC) region—foods typically rich in refined carbohydrates, sugars, and fats that contribute to the growing prevalence of obesity and type 2 diabetes. Focusing on Qatar, the authors develop a culturally specific meal-planning exchange list for 32 commonly consumed GCC and Qatari items. Nutrient data were validated using Food Processor SQL, revealing strong correlations between data sources (r = 0.81–0.85, p < 0.001) and notable variability in macronutrient composition. Fried and syrup-based foods contained higher levels of fats and carbohydrates, while legume- and vegetable-based dishes provided more protein and fiber. The resulting exchange list emphasizes moderation and portion control, offering dietitians a culturally relevant tool to enhance nutrition counseling and support diabetes and weight management across the GCC region.

Comments

Title

It is recommended that acronyms not be used in the title unless they are clearly explained, in order to improve the reader's understanding.

Abstract

The objective of the study should be stated more precisely, specifying the research design, sample size, and period of study. The methodology should be described briefly and clearly, as it is more relevant than the software used. This explanation would help readers better understand the results presented. The conclusions should be directly linked to the objectives and supported by the findings. Overall, the abstract could be improved for greater clarity and coherence.

Introduction

The introduction appropriately presents the importance of food and dietary patterns in Arab culture, particularly traditional sweets. However, the study's objective should be expressed more clearly and concisely at the end of this section.

Materials and Methods

The unit of analysis comprises 32 traditional Qatari dishes. The study objective should not appear in this section but rather at the end of the introduction. The design of the study and the time period in which it was conducted should be specified.

It would also be useful to indicate what proportion of all traditional dishes were included according to the stated inclusion criteria, and whether the analyzed dishes were prepared domestically or purchased commercially.

Some descriptive elements currently included in this section should instead appear in the discussion.

In addition, there is inconsistency regarding the number of dishes—34 are mentioned in section 2.2, whereas 32 were stated earlier. This discrepancy should be clarified. The methodology for data collection should explain how each source contributed: for example, which dishes were obtained from El Obeid's research, which from the Bahrain food composition table, and which from the Eastern Mediterranean Food Information Database (EMFID). It should also be clarified what data were extracted from each database, how the final dataset was created, and which variables were analyzed. The description of the statistical correlation method should specify which parameters or variables were correlated.

Results

Table 1 should indicate the units of measurement for macronutrients or clarify whether the values ​​represent proportions. It should also specify which two variables were correlated. Tables 2 and 3 would benefit from more detailed explanations or captions.

Discussion

The discussion begins by referring again to 34 dishes, although 32 were previously mentioned; this discrepancy should be resolved. In some parts, results are mixed with interpretation — these should be reorganized so that results are described in the previous section, while this section focuses on interpretation, strengths, and weaknesses in light of existing literature. The discussion should also be supported by appropriate references.

Conclusion

The conclusion reads more like a summary than a contribution to knowledge. It should be rewritten to reflect the study's objectives and to highlight its main contributions and implications.

Overall Assessment

This is an interesting and valuable study that provides useful information and could serve as a foundation for improving traditional dietary practices. However, the manuscript requires structural and methodological clarification to enhance its readability and ensure that each section fulfils its intended purpose.

Reviewer 2 Report

Comments and Suggestions for Authors

The study addresses a culturally meaningful and regionally relevant topic; however, its scientific depth and originality remain somewhat limited. The manuscript primarily presents descriptive findings, focusing on correlation analysis without advancing into more comprehensive statistical, mechanistic, or translational exploration. While the development of a culturally adapted exchange list is valuable, the absence of clinical or behavioral validation restricts the practical applicability and generalizability of the results. Additionally, the Discussion section reiterates the results rather than providing new conceptual perspectives or in-depth interpretation. The manuscript would also benefit from a clearer framework illustrating how this exchange system could be integrated into national nutrition policies, digital dietary tools, or public health strategies. Overall, although the topic is of practical interest, the current version does not fully meet the scientific rigor and innovation standards required for publication in Nutrients.